# Trypanosomatid parasite dynamically changes the transcriptome during infection and modifies honey bee physiology

Qiushi Liu[1], Jing Lei [1], Alistair C. Darby[2] & Tatsuhiko Kadowaki [1]*

It is still not understood how honey bee parasite changes the gene expression to adapt to the host environment and how the host simultaneously responds to the parasite infection by modifying its own gene expression. To address this question, we studied a trypanosomatid, *Lotmaria passim*, which can be cultured in medium and inhabit the honey bee hindgut. We found that *L. passim* decreases mRNAs associated with protein translation, glycolysis, detoxification of radical oxygen species, and kinetoplast respiratory chain to adapt to the anaerobic and nutritionally poor honey bee hindgut during the infection. After the long term infection, the host appears to be in poor nutritional status, indicated by the increase and decrease of *take-out* and *vitellogenin* mRNAs, respectively. Simultaneous gene expression profiling of *L. passim* and honey bee during infection by dual RNA-seq provided insight into how both parasite and host modify their gene expressions.

[1] Department of Biological Sciences, Xi'an Jiaotong-Liverpool University, 111 Ren'ai Road, Suzhou Dushu Lake Higher Education Town, Suzhou, Jiangsu 215123, China. [2] Institute of Integrative Biology, University of Liverpool, Liverpool L69 7ZB, UK. *email: Tatsuhiko.Kadowaki@xjtlu.edu.cn

Honey bees (*Apis mellifera*) play a significant role in agricultural crop production and ecosystem maintenance. Nevertheless, the number of managed honey bee colonies has dramatically declined across North America and Europe since 2006[1]. Although there are many potential causes for the observed declines, pathogens/parasites are considered major threats to the health of honey bees[2–4]. There are different kinds of honey bee pathogens/parasites, such as viruses, bacteria, fungi, protozoans, and mites[2]. The honey bee host responses to infections have been previously characterized[5]. For instance, apoptosis seems to be an important response to microsporidian infection and the Toll and Imd immune signaling pathways respond to viral infection. However, the consequences of eliciting and maintaining immune responses on honey bee physiology have not yet been fully understood. Furthermore, we poorly understand how parasites adapt to the honey bee environment when they start establishing the infection and how they react against the host responses during infection maintenance. Elucidating these processes will provide critical insight into understanding honey bee (host)–parasite interactions.

Two *trypanosomatidae* species, *Lotmaria passim* and *Crithidia mellificae*, were shown to infect honey bee. *C. mellificae* was first identified in Australia in 1967[6]. Recently, a novel trypanosomatid parasite was discovered and named *L. passim*[7]. *L. passim* seems to be more prevalent than *C. mellificae*[7–18]. Although the association of *L. passim* infection with winter mortality of honey bee colonies was suggested in several studies[19,20], the effects of *L. passim* infection on honey bee health and colony survival remain poorly understood. *L. passim* can be cultured in medium and specifically infects the hindgut when orally introduced to the honey bee[7]. These characteristics are similar to those of *Crithidia bombi*, a major trypanosomatid parasite of bumble bee[21,22]. *C. bombi* infection dramatically reduces colony-funding success, male production and colony size[23]. Moreover, it has been shown that *C. bombi* infection impairs the ability of bumble bees to utilize floral information[24]. Several genes in immune signaling pathways, such as *Spatzle*, *MyD88*, *Dorsal*, *Defensin*, *Hymenoptaecin*, and *Apidaecin* are also upregulated during the early stage of *C. bombi* infection in bumble bee[22,25,26]. However, changes in *C. bombi* gene expression profile during the infection have never been studied. Since both *L. passim* and *C. bombi* are specifically present in hindguts of honey bee and bumble bee, respectively, they must interact with the gut microbiota. In fact, there is a negative correlation between the infection level of *C. bombi* and the relative abundance of *Apibacter*, *Lactobacillus* Firm-5 and *Gilliamella* bacteria in the gut microbiota of bumble bees[27]. Yet, a recent study has shown that pretreatment with *Snodgrassella alvi* makes normal honey bees more susceptible to *L. passim* infection under natural condition[28]. A relatively simple honey bee gut microbiota shapes the gut microenvironment not only lowering the oxygen but also the pH level[29]. Furthermore, the microbiota utilizes the indigestible compounds of pollen to produce short-chain fatty acids and organic acids which help to maintain the nutritional status of the honey bee[29]. However, it is not known whether *L. passim* affects the functions of the honey bee gut microbiota.

The genome of *L. passim* was sequenced and contains ~9000 protein-coding genes which are transcribed as polycistronic mRNAs, like in other trypanosomatid species[30]. This demonstrates that the amount of mRNA is primarily controlled by post-transcriptional mechanisms such as *trans*-splicing and polyadenylation[31]. It is thus possible to profile the gene expression of *L. passim* during its infection in honey bee by transcriptome analysis. In this study, we characterized host–parasite interactions using *L. passim* and *A. mellifera* as a model system. Cultured *L. passim* cells were orally introduced to newly emerged honey bees and the gene expression profiles of *L. passim* (parasite) and the honey bee (host) were simultaneously analyzed at the early, middle, and late stages of infection. We will discuss how honey bees respond to *L. passim* infection, how their physiological states are modified, and how *L. passim* changes its gene expression to establish and maintain the infection in the host environment.

## Results

**Infection of honey bees by *L. passim*.** We infected newly emerged honey bees with $10^5$ *L. passim* by ingestion and maintained them with 50% (w/v) sucrose under laboratory condition. As shown in Fig. 1a, the number of parasites remained constant up to eight days and then there was dramatic increase at 15 and 22 days after the infection. However, there was large variation in the number of parasites among the infected individual honey bees. These results demonstrate that *L. passim* starts to actively proliferate in the honey bee hindgut between 8 and 15 days after the infection.

We also tested the survival rates of honey bees infected by *L. passim* and fed with 50% (w/v) sucrose at 33 °C under laboratory condition. The infected honey bees survived 26–42 days after infection and this time period was slightly shorter than that of control uninfected honey bees (Fig. 1b). These results indicate that the accumulation of parasites in the honey bee hindgut does not induce rapid death of the host under laboratory condition.

**Effects of *L. passim* infection on honey bee gut microbiota.** Because *L. passim* and the majority of gut microbiota co-exist in the honey bee hindgut, they are likely to interact. To test for the potential interactions, we infected two days old honey bees with *L. passim* and fed them with a mixture of sucrose and pollen under laboratory condition. There were no significant differences in the abundance of universal bacteria and Firmicutes (*Lactobacillus* Firm-4 and Firm-5) with the whole guts of both parasite-infected and uninfected control honey bees at 7 and 15 days after infection (Fig. 1c, d). These results suggest that *L. passim* does not dramatically affect the general landscape of honey bee gut microbiota.

**Changes in the gene expression profile of *L. passim*.** We infected newly emerged honey bees with *L. passim*, returned them to their original hive, and then collected the parasite-infected honey bees at 7, 12, 20, and 27 days after infection (post-infection (PI) 7, 12, 20, and 27). The RNA expression profiles of *L. passim* were analyzed together with those of cultured parasites by RNA-seq using six replicates for each sample. The mapping rates of clean reads to *L. passim* genome were 0.05–52.49% with the lowest and highest average rates of 4.52% and 46.7% at PI 7 and PI 27, respectively. The mapping rate of clean reads derived from the cultured *L. passim* samples was over 87% (Supplementary Data 1). These results were consistent with Fig. 1a showing that parasite population increases 8 days after the infection under laboratory condition. Pairwise comparisons between cultured parasites and honey bee-infecting parasites indicated that 863, 2603, 2531, 2365 genes were upregulated and 1457, 2483, 2322, 2242 genes were downregulated at PI 7, 12, 20, and 27, respectively (Supplementary Data 2). The downregulated genes at each stage of the infection were enriched with many common gene ontology (GO) terms (Supplementary Data 2), suggesting that these samples can be used to identify the differentially expressed genes (DEGs) even though there is heterogeneity in overall transcriptomes of PI 12 and PI 20 samples (Supplementary Fig. 1a). The upregulated genes also shared the same enriched GO terms at each stage of the infection (Supplementary Data 2). Among the upregulated genes, the ones upregulated from the middle to late stage of infection (PI 12–27) represent the highest fraction (1206, 35.8%) without enrichment of the specific GO terms (Fig. 2a and Supplementary Data 5). The second highest fraction corresponded to the genes continuously

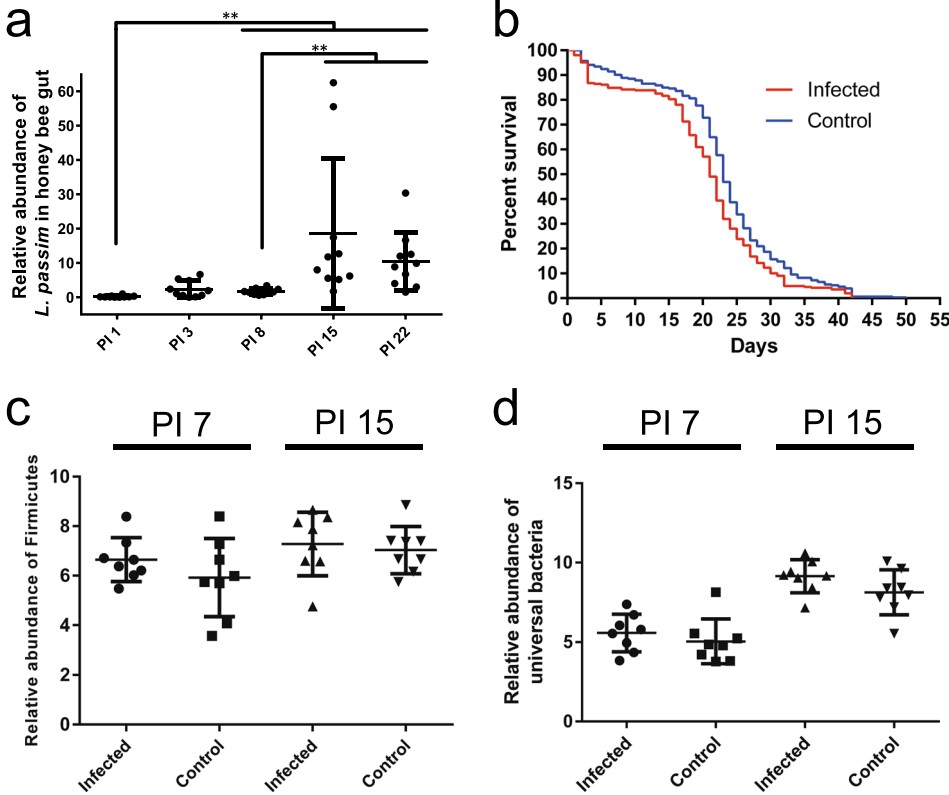

**Fig. 1 The effects of *L. passim* infection on honey bee survival and gut microbiota. a** The abundance of *L. passim* in individual honey bees ($n = 10$) at 1, 3, 8, 15, and 22 days after the infection (PI 1–22). In order to compare the relative abundance of *L. passim*, one honey bee sample on day 1 was set as 1 and thus nine samples were statistically analyzed for day 1. Steel-Dwass method was used for the statistical analysis between the different time points. Mean values ± SD (error bars) are shown. **$P < 0.01$. **b** Survival of honey bees infected by *L. passim* at 33 °C under laboratory condition. Newly emerged honey bees ($n = 96$–110) were either infected by $10^5$ *L. passim* (Infected, red) or fed with sucrose solution (Control, blue) at day 0, and then the survival rates were recorded every day. The experiment was repeated three times and the average values are shown. The data were statistically analyzed by Log-rank (Mantel–Cox) test ($P = 0.0002$). Relative abundance of Firmicutes (**c**) and universal bacteria (**d**) in *L. passim*-infected (Infected) and the uninfected (Control) honey bees at PI 7 and PI 15 ($n = 8$ for each time point). Mean values ± SD (error bars) are shown. Unpaired *t* test (two-tailed) was used for the statistical analysis.

upregulated throughout the parasite infection (575, 17.1%) with enrichment of GO terms, for example, "rRNA processing" and "proteolysis" (Fig. 2a and Supplementary Data 3). They included many genes encoding RNA processing/export-associated proteins and peptidases as well as *L. passim* homolog of glycoprotein (GP) 63 (leishmanolysin) (Supplementary Data 3). In the downregulated genes, the ones continuously downregulated throughout the parasite infection represent the highest fraction (948, 30.1%) and were enriched with many GO terms including "structural constituent of ribosome", "axoneme", "oxidoreductase complex", "glycolytic process", and "entry into host". In fact, many genes encoding ribosomal proteins, paraflagellar rod proteins, tryparedoxin, tryparedoxin peroxidase, glycolytic enzymes were present (Fig. 2b and Supplementary Data 4). Multiple *Amastin* genes should represent the GO term "entry into host". They are cell-surface proteins abundantly expressed in the intracellular amastigote stages of *Trypanosoma* spp. and *Leishmania* spp[32,33]. The second highest fraction (759, 24.1%) corresponded to the genes downregulated from the middle to late stage of infection (PI 12–27) and many of them encoded proteins in the kinetoplast respiratory chain with enriched GO terms related to "ATP synthesis coupled proton transport" (Fig. 2b and Supplementary Data 6). These results demonstrated that *L. passim* dynamically changes the transcriptome to establish and maintain hindgut infection. Identification of DEGs followed by the GO term enrichment analysis between the sequential stages of *L. passim* infection demonstrated that proteolytic activity and GP63 may

further increase from PI 7 to PI 12 (Supplementary Data 7). Meanwhile, mRNAs encoding RNA processing/export-associated proteins decreased at the same time, suggesting that their expression was highest at PI 7 then fell at later stage of the infection (Supplementary Data 7). This is consistent with that GO terms related to RNA processing/export were only enriched with the upregulated genes at PI 7 but not other stages (Supplementary Data 1). Nevertheless, the amounts of these mRNAs should be higher than those in the cultured parasites at any stage of the infection (Supplementary Data 3). Down-regulation of many genes encoding proteins in the kinetoplast respiratory chain from PI 7 to PI 12 is consistent with above result that they were enriched in the downregulated genes during PI 12–27 (Supplementary Data 6).

**Changes in the gene expression profile of honey bee hindgut.**
We next compared the gene expression profiles between *L. passim*-infected and uninfected control honey bee hindguts at the same time points (PI 7–27) by RNA-seq using six replicates for each sample. The mapping rates of clean reads derived from 24 control honey bees to honey bee genome were over 93% except two samples (48.55% for C-20-B and 70.33% for C-27-E) and those derived from 24 *L. passim*-infected honey bees were 37–96% (Supplementary Data 1). The mapping rates were highest at PI 7, the time point with the lowest level of *L. passim* infection. Clustering analysis revealed that the overall transcriptomes of the control honey bee hindguts were similar to those of the *L. passim*-infected ones irrespective of

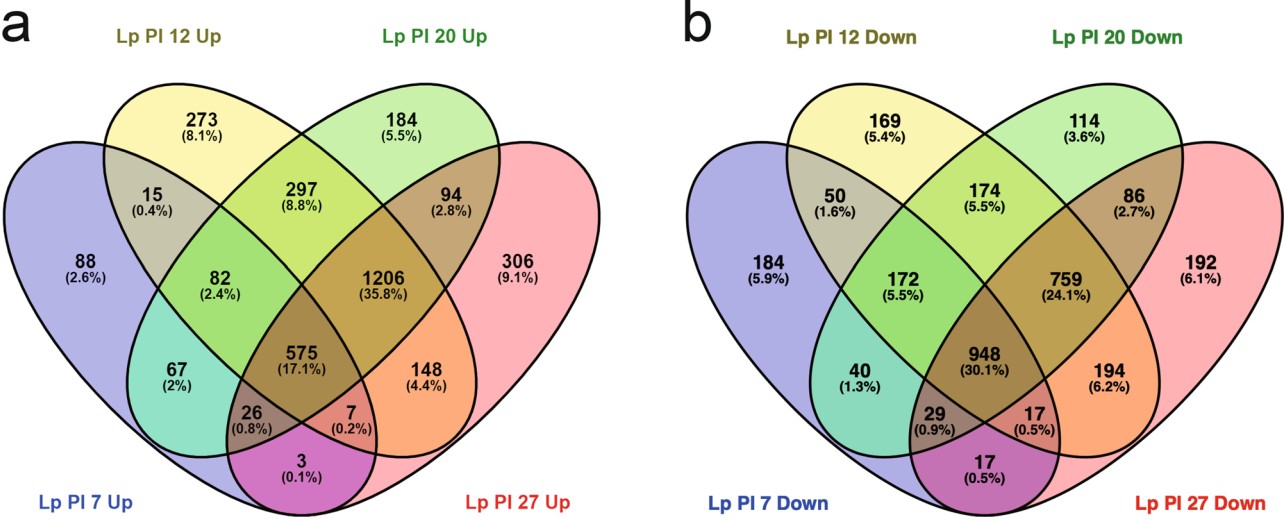

**Fig. 2 Global transcriptomic profiles of *L. passim* at the different time points of infection in the honey bee hindgut.** Venn diagram to indicate *L. passim* genes upregulated (**a**) or downregulated (**b**) at four different time points (PI 7–27) of the infection.

the stage of parasite infection (Supplementary Fig. 1b). These results suggest that gene expression of the honey bee hindgut does not change dramatically upon *L. passim* infection. Pair-wise comparisons between the age-matched control and infected honey bees indicated that 21, 112, 70, and 137 genes were upregulated and 15, 146, 134, and 26 genes were downregulated at PI 7, 12, 20, and 27, respectively. Among them, only 146 genes downregulated at PI 12 showed the enrichment of GO terms related to "nucleosome" (Supplementary Data 8). Very few genes were shared between different infection time points (Fig. 3a, b); however, ten genes upregulated at both PI 20 and PI 27 contained nutrition and starvation-related *take-out-like carrier protein* (*TO*) and *facilitated trehalose transporter Tret1-like* (*Tret1*) genes (Table 1). These results suggest that *L. passim*-infected honey bees are likely to be starved and that the gut epithelial cells are in poor nutritional status at the late stage of infection.

***Vitellogenin (Vg)* mRNA in *L. passim*-infected honey bees.** *Vg* mRNA level was shown to be correlated with the nutritional status of worker honey bee[34]. Enhanced expression of *TO* and *Tret1* mRNAs in the hindguts of *L. passim*-infected honey bees led us to test *Vg* mRNA levels in their fat bodies. As shown in Fig. 4, *Vg* mRNA levels in the fat bodies of honey bees at 20 days after *L. passim* infection were significantly lower than those of uninfected controls. These results suggest that *L. passim*-infected honey bees are in poor nutritional status.

## Discussion
We found that ingested *L. passim* reaches the honey bee hindgut and colonizes it as previously reported[7]. The number of parasites dramatically increases around 7–12 days and then reaches plateau at 20 days after infection with $10^5$ cells under both laboratory and hive conditions. Accumulation of the parasites in the hindgut slightly decreases the honey bee survival but never results in rapid death. This appears to be the case under hive condition as well, since we recovered *L. passim*-infected honey bees at 37 days after infection. Many parasites share these infection characteristics in order to increase dissemination to other individuals through feces[6,35–37].

We did not find evidence for interactions between the honey bee gut microbiota (Firmicutes and universal bacteria) and *L. passim*. The number of gut bacteria was not affected by *L. passim* infection and similar observation had been previously made[38].

Interestingly, higher level of *L. passim* infection increased the number of *Gilliamella apicola* and universal bacteria in honey bee hindgut under hive but not under laboratory condition[28]. This may suggest that a small number of *L. passim* modifies the honey bee hindgut environment and stimulates colonization of gut microbiota but high degree of modification by a large number of parasites does not support further colonization. Lack of interaction between *L. passim* and honey bee gut microbiota is also consistent with the recent studies that show bumble bee microbiota is unaffected by *C. bombi* infection[39,40]. Nevertheless, whether *L. passim* affects the number of other bacterial species that were not examined in this study and/or the functions of the gut microbiota - rather than just the number - remains to be tested.

We found that *L. passim* dramatically changes its gene expression profile throughout the different stages of infection in honey bee hindgut. Many ribosomal protein-coding genes are downregulated throughout the infection period. The nutritional status of the honey bee hindgut must be poor compared to that of culture medium because most of the nutrients derived from pollen and honey (food) are absorbed in the midgut and only residual compounds of food and metabolites from the gut microbiota are available. Thus, limited supply of nutrients may down-regulate the ribosomal protein-coding genes to suppress protein translation. Similarly, lack of sufficient glucose in the hindgut may also result in decreasing mRNAs for glycolytic enzymes.

Interestingly, many genes associated with RNA processing/export were continuously upregulated in the parasites during the infection of hindgut particularly at the early stage (PI 7). Stimulating, for example, *trans*-splicing and polyadenylation of polycistronic mRNAs would be necessary to maintain the proliferation of parasites in the honey bee hindgut even though ribosome biogenesis and energy (ATP) synthesis are reduced. Increase of the transcripts for various proteases is also noteworthy. Some of these proteases could be secreted and may play roles in modifying the tissue to facilitate growth in the honey bee hindgut.

Anaerobic environment of the honey bee hindgut appears to have dramatic effect on *L. passim* gene expression and energy metabolism as well. *L. passim* was initially cultured under normoxia and migrated into the anaerobic honey bee hindgut, causing it to down-regulate the genes involved in the kinetoplast respiratory chain during middle to late stages of the infection (PI 12–27). Similar shifts in energy metabolism are also observed in

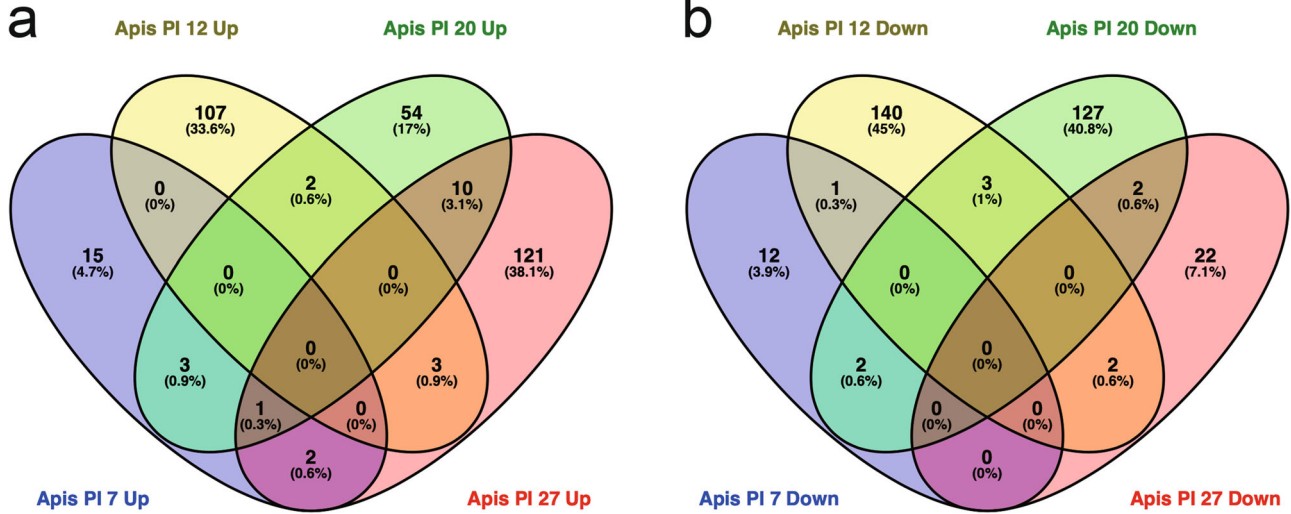

**Fig. 3 Global transcriptomic profiles of honey bee hindguts at the different time points of *L. passim* infection.** Venn diagram to indicate honey bee genes upregulated (**a**) or downregulated (**b**) in response to *L. passim* infection at four different time points (PI 7–27).

**Table 1 List of *A. mellifera* genes upregulated at both PI 20 and PI 27.**

| Gene ID | Annotation |
|---|---|
| NP_001011640.1 | Take-out-like carrier protein precursor |
| XP_623950.2 | Facilitated trehalose transporter Tret1 |
| XP_623221.2 | Elongation of very long-chain fatty acids protein AAEL008004 |
| XP_394861.4 | O-acyltransferase like protein isoform X1 |
| XP_001120006.2 | Protein lethal (2) essential for life |
| XP_395671.3 | Probable cytochrome P450 6a17 |
| XP_397283.1 | ATP-binding cassette sub-family G member 5 |
| XP_001120450.1 | E3 ubiquitin-protein ligase DCST1 isoform X1 |
| XP_026298733.1 | Uncharacterized protein LOC409465 |
| XP_001119879.1 | Uncharacterized protein LOC724580 |

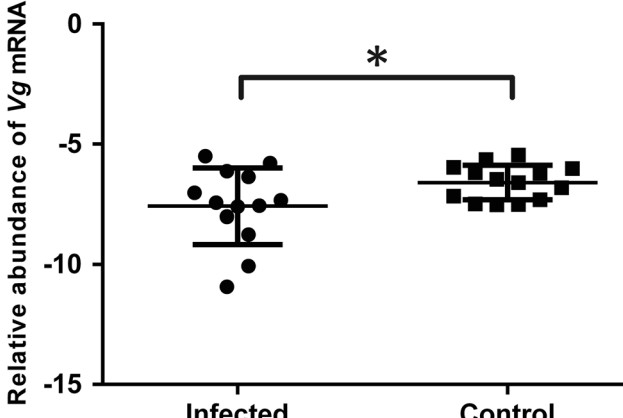

**Fig. 4 Relative amounts of *Vg* mRNA in the fat bodies of *L. passim*-infected and uninfected honey bees.** Relative amounts of *Vg* mRNA in the fat bodies of *L. passim*-infected (Infected, $n = 13$) and uninfected honey bees (Control, $n = 14$) were measured at 20 days after the infection under hive condition. The samples were obtained by two independent experiments. Mean values ± SD (error bars) are shown. Unpaired $t$ test (two-tailed) was used for the statistical analysis ($*P < 0.05$).

other trypanosomatid parasites[41–44]. The parasites actively swim in the cultured medium using the flagellum and ATP; however, the loss of flagellum (see below) makes them immobile in the honey bee hindgut. Thus, the reduced demand for ATP may also be involved in down-regulating glycolysis and kinetoplast respiratory chain. Furthermore, the continuous down-regulation of *tryparedoxin 1*, *tryparedoxin-like*, and *tryparedoxin peroxidase* genes throughout the infection cycle is also consistent with the reduced production of radical oxygen species in anaerobic environment. Thus, *L. passim* appears to shift its growth and energy metabolism depending on the poor nutritional and anaerobic conditions of the honey bee hindgut.

In terms of morphology, *L. passim* grown in culture medium contains a flagellum. However, the flagellum is absent when the parasites accumulate in the honey bee hindgut. This is also supported by the decrease of mRNAs for proteins involved in flagellar formation such as paraflagellar rod proteins[45]. *L. passim* appears to change its cell surface proteins during the infection cycle. Similar to *C. bombi* and *Crithidia expoeki*[46], multiple *Amastin* genes appear to exist in *L. passim* genome and some are upregulated at PI 12–27 (Supplementary Data 1); however, many are downregulated and *GP63* is instead upregulated throughout the infection cycle. *Leishmania* GP63 was proposed to be essential for the promastigotes to attach to the insect gut wall[47–49]. *Leptomonas* GP63 was suggested to have the same function[50], which may also apply to *L. passim* GP63. Thus, switching Amastin to GP63 on the cell surface could be critical to establish and maintain the infection of *L. passim* in the honey bee hindgut.

The honey bee hindgut response to *L. passim* infection is characterized by changes in gene expression. However, the number of DEGs is relatively small and very few of them are shared at different infection time points. This could be due to the large heterogeneity in individual worker's gene expression as well as physiological state associated with performing different task even at the same age in the colony. Previous studies showed *C. bombi* infection induces immune responses in bumble bees, for example, increase of transcripts for antimicrobial peptides (AMPs)[25,26]. These bumble bee AMPs were shown to directly inhibit the growth of *C. bombi*[51,52]. We found that only one AMP, Apidaecin type 14, was upregulated by *L. passim* infection in the honey bee hindgut at PI 12 (Supplementary Data 8). Based on the metabolic pathways identified by genome sequencing, *L. passim* should be able to synthesize β-1,3-glucan on its cell

surface, and beta-1,3-glucan binding protein 1, which was upregulated in the honey bee hindgut at PI 27 (Supplementary Data 8), could bind to *L. passim* through glucan moiety[53]. This would result in the activation of prophenoloxidase (pro-PO) (for melanin deposit) or Toll signaling pathway[54–57] to defend against the parasite infection. Nevertheless, honey bee does not appear to elicit strong immune responses against *L. passim* infection as previously reported[58]. TO was shown to be involved in controlling the feeding behavior and nutritional status of fruit fly by binding juvenile hormone and the mRNA expression is highly stimulated in the head and gut tissues by starvation[59,60]. Thus, up-regulation of *TO* in the hindgut at PI 20–27 suggests that *L. passim*-infected honey bees were under starvation. Trehalose level in the hemolymph is reduced in the fruit fly under starvation[61] and this is consistent with the simultaneous increase of *Tret1* mRNA in the hindgut of *L. passim*-infected honey bees. The decrease in hemolymph trehalose level is caused by its reduced release from fat body and increased uptake by various tissues. Accordingly, we also found *Vg* mRNA is reduced in the fat bodies of *L. passim*-infected honey bees compared to the uninfected controls. Honey bee gut microbiota was shown to produce various metabolites such as organic acids, some of which would be absorbed by the host and affect weight gain as well as appetite behavior[29]. *L. passim* may suppress this process and lead the honey bee into poor nutritional status. Furthermore, increase of transcripts for elongation of very long-chain fatty acids protein AAEL008004 and ATP-binding cassette sub-family G member 5 suggests that phospholipid biosynthesis and sterol transport in the hindgut epithelial cells may be stimulated. Up-regulation of *protein lethal (2) essential for life* (a member of *HSP20* family) gene may also help to refold denatured proteins in the cells stressed by *L. passim* infection (Table 1).

Our results obtained by dual RNA-seq show how *L. passim* modifies its gene expression to adapt to the honey bee hindgut environment and, at the same time, how the honey bee modifies its gene expression against the *L. passim* infection. Further study on the roles of these genes is necessary. Honey bee-*L. passim* should represent a good model system to understand host–parasite interactions at three different layers, cell, organism, and population (colony).

## Methods

**Culture of *L. passim* and the oral infection of honey bee**. *L. passim* was grown in modified FPFB medium[62] supplemented with 10 kU/mL penicillin (Beyotime), 10 mg/mL streptomycin (Beyotime), and 50 mg/mL gentamycin (Noble Ryder) at 25 °C. The cells were collected during logarithmic growth phase and washed once with PBS followed by resuspension in sterile 10% sucrose/PBS at 20,000 viable cells/μL. Newly emerged honey bee workers were collected and then starved for 2–3 h. Honey bees were divided into two groups: One group was fed with 5 μL of 50% sucrose solution (Control). The other group was fed with 5 μL of 50% sucrose solution containing *L. passim* (100,000 cells in total). After the oral infection, the honey bees were marked with oil paint on the thorax and returned to the hive (natural condition). To test under laboratory condition, the control and infected honey bees were maintained in metal cages at 33 °C.

**Quantitative PCR (qPCR) to measure *L. passim***. We sampled 10 honey bees at 1, 3, 8, 15, and 22 days after the infection under the laboratory condition and extracted the genomic DNAs from the whole abdomens of individual bees using DNAzol® reagent (Thermo Fisher). To quantify *L. passim*, the part of *internal transcript spacer region 2 (ITS2)* in *ribosomal RNA (rRNA)* gene was amplified for qPCR (Forward primer: GGGTCTTTTGTGATCGGGATAA; Reverse primer: CAAAAAGATGCCTAACGTGAAGAA). Honey bee *AmHsTRPA* was used as the reference (Forward primer: TAGCGTACATGTGGTGCTGT; Reverse primer: GCTAGGCTCCACGTAATCCA). The relative abundances of *L. passim* in the individual honey bees were calculated by $\Delta C_t$ method. Steel-Dwass method was used for the statistical analysis.

**Survival rates of *L. passim*-infected honey bees**. To test the effect of *L. passim* infection on the survival of honey bee, we maintained 100 *L. passim*-infected and 100 control honey bees in the separate metal cages and fed them with 50% sucrose

solution at 33 °C. The dead bees were removed from the cages and counted every day. The experiment was repeated three times. The results were statistically analyzed by Log-rank (Mantel–Cox) test. In order to confirm *L. passim* infection, we sampled the dead bees (one from the control group and six from the infected group) at 15 days after the infection. Genomic DNA was extracted from the individual bees and the parasite was detected by PCR targeting to the part of *ITS1* in *rRNA* gene (Forward primer: GCTGTAGGTGAACCTGCAGCAGCTGGATCATT; Reverse primer: GGAAGCCAAGTCATCCATC)[63]. Honey bee *AmHsTRPA* was used as the reference (Forward primer: CACGACATTCAAGGTTTAAGAAATCACG; Reverse primer: TCAGTTATTCTTTTCCTTTGCCAGATTT)[64]. Six dead bees from the infected group were specifically positive for *L. passim*.

**Quantification of gut microbiota in honey bee**. We maintained newly emerged honey bees with the hive frame for two days to allow them to acquire the core gut microbiota. The honey bees were then divided to *L. passim*-infected and the control groups as above and fed with 50% sucrose solution with pollen. We collected eight bees from each group at 7 and 15 days after the infection. Bee abdomens were washed with 75% ethanol followed by immersion in sterilized PBS. We then dissected the whole gut and extracted the genomic DNA from individual bee. Universal bacteria (all bacterial species) (Forward primer: AGAGTTTGATCCTGGCTCAG; Reverse primer: CTGCTGCCTCCCGTAGGAGT) and phylum-specific *Firmicutes* (Forward primer: TGAAACTYAAAGGAATTGACG; Reverse primer: ACCATGCACCACCTGTC) were quantified by qPCR[65,66]. Honey bee *β-actin* was used as the reference (Forward primer: TTGTATGCCAACACTGTCCTTT; Reverse primer: TGGCGCGATGATCTTAATTT)[67] and the statistical analysis was carried out by unpaired *t* test (two-tailed).

**Preparation of RNA-seq samples**. We sampled *L. passim*-infected and the control honey bees from the hive at 7, 12, 20, and 27 days after the infection. Total RNA samples were extracted from the hindguts of 12 individual honey bees for each time point. The level of *L. passim* infection was determined by quantifying the *18S rRNA* by qRT-PCR (Forward primer: GAAAGGAACCACTCCCGTGT; Reverse primer: GTCCCGTCCATGTCGGATTT) and honey bee *18S rRNA* was used as the reference (Forward primer: ACCACATCCAAGGAAGGCAG; Reverse primer: ACTCATTCCGATTACGGGGC)[68] since it is one of the most abundant and thus stable RNA in the cells[69]. Six samples with similar levels of *L. passim* infection were analyzed by RNA-seq at each time point. We also prepared six samples of total RNA from cultured *L. passim* at the logarithmic growth phase. All samples were sequenced by BGISEQ-500 platform at the Beijing Genomics Institute. At least 6 GB of clean data were obtained from each sample and thus analysis of very rare transcripts may not be reliable. There are 6 samples of cultured *L. passim*, 24 samples of the parasite-infected honey bees, and 24 samples of the control uninfected honey bees.

**Annotation of *L. passim* protein-coding genes**. Since genome sequence of *L. passim* (strain SF, GenBank number: AHIJ00000000.1) in NCBI does not contain information for the gene annotation, we annotated *L. passim* genes first. The clean reads of RNA-seq mapped to *L. passim* genome by TopHat V2.1.0[70] were assembled using Cufflinks v2.2.1[71]. We then collected the assembled transcripts from all RNA-seq samples using Cuffmerge[71]. Gene annotation for the collected *L. passim* transcript and genome sequences was conducted by parasite genome annotation pipeline companion with *Leishmania major Friedlin* as the reference organism[72]. The gene-annotated GFF3 file and the protein-coding sequences were used for the bioinformatic analyses.

**Mapping of RNA-seq reads to honey bee and *L. passim* genomes**. Honey bee reference genome, annotation GFF, and protein fasta files were downloaded from NCBI. We used a gffread program (http://ccb.jhu.edu/software/stringtie/gff.shtml) to convert the honey bee GFF to GTF file, and then extracted the splice junctions and exon positions from the converted GTF file using HISAT2 (Version: 2.1.0)[73]. The clean reads of RNA-seq were mapped to the HISAT2 index and the SAM files were converted and sorted by SAMtools[74]. We used HTSeq-count to count the number of reads mapped to each gene in the sorted SAM file[75]. The HISAT2 index for *L. passim* genome was similarly built using the gene-annotated GFF3 file and the protein-coding sequences prepared as above.

**Identification of DEGs for both *L. passim* and honey bee**. We analyzed raw read counts from both *L. passim* and honey bee in RStudio (Version 1.1.383) using the generalized linear model-based method of edgeR package (Version 3.24.2) from Bioconductor[76]. Normalization was performed by Trimmed Mean of M-values implemented in the edgeR[77]. We set up a filter to remove any reads with less than one count per million mapped reads. After testing, the Benjamini–Hochberg method was applied to control the false discovery rate (FDR) across the detected loci. For *L. passim*, we corrected the raw *P* value for each gene and identified the DEGs between cultured parasites and honey bee-infecting parasites at PI 7, 12, 20, and 27 by the threshold of FDR < 0.01. For honey bee, the DEGs between *L. passim*-infected and the control honey bee hindguts at the same time points (PI 7–27) were identified by the threshold of FDR < 0.05. After identifying the DEGs,

we used Venny (http://bioinfogp.cnb.csic.es/tools/venny/index.html) to build the Venn diagrams to summarize the relationships between the DEGs.

**GO term enrichment analysis (Fisher's exact test)**. We first constructed the local blast databases using OmcisBox 1.2.4 software with the taxonomy IDs (https://www.biobam.com/omicsbox/). In total, 446,413 and 6,225,195 protein sequences of Kinetoplastida (Taxonomy ID: 5653) and Hexapoda (Taxonomy ID: 6960) were downloaded respectively from NCBI with the GI numbers. We then built a local BLAST database using two FASTA files with above protein sequences in the OmicsBox. The lists of DEGs for *L. passim* and honey bee identified above were individually analyzed. We run BLASTP (parameters: E-value hit filter = 1e$^{-10}$; Number of blast hits: 20) against the appropriate local BLAST database created above. For each dataset, InterPro scan was simultaneously run using EMBL-EBI public web-service database with the default settings. After the annotation, we merged the InterPro scan GO terms, and then performed the enrichment analysis (Fisher's exact test) using the program embedded in OmicsBox. The entire protein-coding sequences of *L. passim* and *A. mellifera* were used to generate the reference annotation lists. After obtaining the enriched GO terms, we identified the most specific ones by the cut-off value of FDR < 0.05.

**Quantification of *Vg* mRNA by qRT-PCR**. We sampled *L. passim*-infected and the control honey bees from the hive at 20 days after the infection. Fat bodies were dissected from the individual bees, and then total RNA was extracted as described above. We measured the relative amount of *Vg* mRNA to *18S rRNA* by qRT-PCR (Forward primer: GTTGGAGAGCAACATGCAGA; Reverse primer: TCGATC-CATTCCTTGATGGT)[78] using $\Delta C_t$ method[28]. The results were statistically analyzed by unpaired *t* test (two-tailed).

**Statistics and reproducibility**. Statistical analyses were performed on GraphPad Prism 7 and no data point was excluded. All data presented are from representative independent experiments. The biological replicates and applied statistical tests are described in the respective parts of Methods section as well as figure legends. Significant differences were considered with *P*- value of <0.05(*) and <0.01(**).

**Reporting summary**. Further information on research design is available in the Nature Research Reporting Summary linked to this article.

## Data availability

The RNA-seq data are available at the SRA database with accession number PRJNA587465. All other data are available from the corresponding author on reasonable request.

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

## Acknowledgements

We thank Yichen Lu and Xinyi Li for their contribution to conduct some of the experiments. We are grateful to Dr. Duo Peng at Harvard T.H. Chan School of Public Health for suggestion on the bioinformatics analysis. We are also grateful to Dr. Michelle Flenniken at Montana State University for suggestion on the experiments. This work was supported by Jinji Lake Double Hundred Talents Programme and XJTLU Research Development Fund (RDF-14-01-11) to T.K.

## Author contributions

A.C.D. and T.K. conceived and designed research strategy. Q.L. and J.L. performed the experiments. Q.L., J.L. and T.K. interpreted the results. T.K. and Q.L. drafted, edited, and revised the paper.

## Competing interests

The authors declare no competing interests.
