## [Peer Review File · Communications Biology]

Reviewers' comments:

Reviewer #1 (Remarks to the Author):

This manuscript describes the host-parasite interactions at the molecular level between the adult honey bee and the trypanosomatid parasite *Lotmaria passim*. It covers both the gene expression of the parasite and the host, but also host survival and changes in host gut microbiota.

To my knowledge no such comprehensive study on the host-parasite interactions between this parasite and its host has ever been performed. There are some point that needs to be addressed.

Line 44: The decline of number of bee hives: please use a references, for instance Pots et al., 2010 Journal of Apicultural Research.

Line 47 and 49: References Evans and Schwarz 2011 a and b are exactly the same (see reference list).

Line 115: do not mention the temperature here as it causes confusion (looks as if the bees were infected at 33°C; I think you mean that bees were kept at 33°C).

Lines 294-298: The paper of Blanchette et al., 1999 refers to *Leishmania* inside the macrophages, which is a situation completely different from the extracellular development of *Lotmaria* in bees. Only comparison with the promastigotes in the sandfly alimentary tract seems valid to me. So whole this paragraph is much too speculative.

Lines 322: To my knowledge, gases like O₂ and CO₂ can easily cross the membranes of cells. This happens for instance in the capillaries of alveoli (lungs) and peripheral tissues of mammals, where oxygen easily crosses the endothelial cells. I would be very surprised if tight junctions between epithelial cells would have any influence on oxygen uptake. Moreover, I do not fully understand your statement that *L. passim* would inhibit the oxygen supply from the basolateral side.

Line 339: Please explain better why the '*L. passim*-honey bee' model is such an interesting model. In fact, I see many parallels with the '*Paenibacillus* larvae-honey bee larvae' model: the pathogen can be cultured in vitro, can be easily administered, gene expression studies of the host (Cornman et al., 2013 PLOS ONE) and the pathogen (De Smet et al., 2014 PLOS ONE) have been studied. In fact, in the case of American foulbrood clinical signs are more obvious, effects on survival more dramatic and many virulence genes are known. Why would the *Lotmaria*-bee model be superior to this one?

Line 386: universal bacteria: please explain what this is.

Line 393: I regret that only one reference gene was used for the qRT-PCR. In fact, you should have determined how many reference genes are required and which are the most stable ones, as done for the first time in honey bee research by Scharlaken et al., 2008 Journal of Insect Science.

Reviewer #2 (Remarks to the Author):

- Brief summary of the manuscript

- Parasite-host interaction between trypanosomatid *Lotmaria passim* and its host honey bee *Apis mellifera* was evaluated by analyzing the expression profiles of both, the parasite and the host, at early, middle, and late stages of infection. Newly emerged bees were experimentally infected with *L. passim* and the effects of infection on the honey bee were monitored under both laboratory and hive conditions. Bees were collected at days 7, 12, 20, and 27 post infection in order to assess dynamic changes in gene expression profile of both, the parasite and the host during the host hindgut infection. The effects of *L. passim* infection on honey bee mortality and gut microbiota were also assessed.

- Overall impression of the work

- Very important work with interesting results as the trypanosomatid *Lotmaria passim* has been recently described and the impact of *L. passim* infection on honey bee health and colony survival is poorly understood. Modern techniques and novel methodological approach were applied in this study providing an important insight into the host-parasite interactions at molecular level. Furthermore, the methodology introduced here is generally applicable in any other research where host-parasite relationship needs to be investigated. The Introduction and the Discussion sections are extraordinary well written. However, Material & Methods and the Result sections seem to have been written by inexperienced researcher and must be significantly corrected (adapted for publishing in scientific paper).

- Specific comments, with recommendations for addressing each comment

- I strongly suggest mentioning the overall number of analyzed genes in both the Abstract and the M&M section. In Supplementary Table 11 there are data for only 22 genes although the obtained results indicate that much more genes were included in the study. I also suggest authors to provide the complete list of all analyzed genes (together with primers used) in one Table. There are 31 specific comments/suggestions I inserted directly in MS in order to help authors to correct the MS.

The paper meets four general criteria:

- Provides strong evidence for its conclusions.
- Novelty
- It is of extreme importance to scientists in the specific field
- It is interesting to researchers in other related disciplines.

♣ Key results:

The results of the study revealed that:

- during *Lotmaria passim* infection of honey bee hindgut, the expression of *L. passim* genes associated with protein translation and the electron transport dynamically changes 'to adapt to the anaerobic and nutritionally poor honey bee hindgut at early stages of infection, and to become dormant at late stages of infection'.
- several genes are continuously up- or down-regulated during infection, including GP63 as well as genes coding for host cell signaling pathway modulators (up-regulated), and those involved in detoxification of radical oxygen species as well as flagellar formation (down-regulated).
- *L. passim* infection only slightly increases honey bee mortality and does not affect the number of microorganisms in the gut microbiota;
- *L. passim* infection induces honey bee innate immune response based on recorded changes in host gene expression. However, the number of differentially expressed genes (DEGs) is relatively small and very few of them are shared at different infection time points. Upon infection, the host appears to be in poor nutritional status, indicated by the increase in the levels of mRNAs for take-out and facilitated trehalose transporter and the decrease of vitellogenin mRNA level.
- simultaneous gene expression profiling of *L. passim* and honey bee during infection provides insight into how both parasite and host modify their gene expressions. This methodological approach is generally applicable in any research investigating host-parasite relationship.

♣ Validity: The manuscript DOES NOT HAVE flaws which should prohibit its publication.

♣ Originality and significance: I feel that the results presented are of immediate interest to many people in the discipline related to honey bees and their parasites, but also to people from other disciplines where host-parasite relationship needs to be investigated.

♣ Data & methodology: Please comment on the validity of the approach, quality of the data and quality of presentation. Please note that we expect our reviewers to review all data, including any extended data and supplementary information. Is the reporting of data and methodology sufficiently detailed and transparent to enable reproducing the results?

- In spite of high-quality data obtained, the presentation does not meet the criteria expected for scientific journal. In fact, Material & Methods and the Result sections seem to have been written by inexperienced researcher and must be significantly corrected (adapted for publishing in scientific paper). Reporting of data and methodology is sufficiently detailed, but the writing style is not adequate.

Overall number of analyzed genes is not clear, so I strongly suggest mentioning that number in both the Abstract and the M&M section. In Supplementary Table 11 there are data for only 22 genes although the obtained results indicate that much more genes were included in the study (e.g. see the sentence: 'Pair-wise comparisons between cultured parasites and honey bee-infecting parasites indicated that 961, 415, 582 genes were up-regulated and 451, 1069, 658, 832 genes were down-regulated at PI 7, 12, 20, and 27, respectively' lines 142-144 in MS as well as Supplementary Tables 1-9). Thus, I suggest authors to provide the list of all analysed genes (together with primers used). Generally, Figures and Supplementary Tables are good and informative.

♣ Appropriate use of statistics and treatment of uncertainties:

- Statistical tests seem appropriate, but that should be confirmed by the person adequately qualified to assess the statistics.

♣ Conclusions:

- The conclusions and data interpretation are robust, valid and reliable.

♣ References:

- The manuscript reference previous literature appropriately. Nevertheless, some additional references should be included as I indicated directly in MS.

♣ Clarity and context:

- The abstract is clear and accessible, and the Introduction and conclusions appropriate. Discussion is the best part of the MS. Sections M&M and Results must be corrected (in accordance to my comments/suggestions inserted directly in MS).

♣ Please indicate any particular part of the manuscript, data, or analyses that you feel is outside the scope of your expertise, or that you were unable to assess fully.

- I feel that I am not able to assess Statistic analysis fully.

Reviewer #3 (Remarks to the Author):

In this study, Liu et al. approach an interesting host-parasite system (the honeybee *Apis mellifera* and a trypanosome gut parasite) trying to understand the intricate array of interactions between the two organisms from multiple points of view. I much appreciate the comprehensive approach, aimed at considering the effect of the parasite on host survival, on the microbial flora associated with the host gut and host gene expression over time during the infection. Furthermore, the authors planned to look at gene expression in the parasite too, to see whether it follows the fluctuations observed in the host. This is quite a novel and challenging approach that must be commended. However, probably for the high challenges associated with such a complex and multi-faceted study, the authors fail in my opinion to provide any strong evidence for any of these aspects.

My major concern is about the design of the RNAseq studies. Sample size is too low there – only two samples for experimental group – and this cannot be accepted as it doesn't really provide any robust scientific significance. Three would be the minimum to calculate any sort of average/median of gene expression data, and even this would be too low when considering individual biological replicates that might vary for so many factors. Nowadays for individual replicates, normally scientist aim to have at least 6-8 samples per group and this is so far from what the authors have here. In FIG. 3A for example, when the two samples in the infPI12 group are so much separated in the PCA analysis, what can be inferred? Furthermore, also the coverage for the RNAseq approach is so low (4GB) compared to what is normally the case these days (20GB or more) that it makes it difficult to consider these results as robust and significant.

Even though sample size is not a major issue for the other analyses (as it is much more reasonable than in the RNAseq study) the authors are not always clear about it through the manuscript. The reader can infer it from the figures of Vg expression, for example, or the quantification of L/ passim and bacteria...but it would help if authors could clearly state it in the format N = XX in the text (results for examples) and in the figure or figure legend. For mortality, sample size is huge but this doesn't appear from the main text – I realised it only after reading the reporting summary.

There are also issues associated with the presentation of the gene expression data and the description of the approaches of analysis that makes it a bit confusing to understand what the authors have done – see points below.

On a positive note, the paper is well written overall, indicating a deep knowledge of the field and thorough research of the literature. The discussion is very comprehensive and the authors have made a good effort to analyse the gene expression results in the context of the biology of the host-parasite system and the previous literature on the same system or similar host-parasite associations. It is unfortunate that the RNAseq study is so weak, and the rest of the results, though possibly interesting, disappear through the manuscript as so much emphasis is given to the gene expression analyses.

Minor comments:

line 76: rephrase

line 77: Drosal = Dorsal

lines 137-138: this is a confusing sentence. Who are "these" and what are the "above results"? please clarify

lines 141-142: what does "and it was an outgroup to the other time points" mean?

FIG. 2&3 C and D: the way how gene expression results are presented in these sections of the two figures and also in the associated main text is a very confusing. Why did the authors decided to present gene expression data with convoluted overlapping analyses represented by confusing Venn Diagrams? I understand the overlaps to identify common genes across time points – but this could have been simply added to the supp mat and discussed in the result section. As it is a time-course study, it would be more indicative to show gene expression data with line graphs indicating numbers

of genes up or down-regulated that go increase or decrease over time in the host and in the parasite. also, what is the up or down-regulation referred to? in the host, my assumption would be that the comparisons are done between infected and control individuals at each time point - so up/down refer to patterns in infected bees with respect to controls...is this true? It's impossible to understand this from the results or the methods. And what about the parasite? are they all compared to parasites in culture? A or B or together? or did the authors followed a completely different approach? this must be explained.

lines 169-173: this should be stated much earlier in the manuscript to provide a rationale for these comparisons...which otherwise are difficult to understand at first for people who are not expert in the field

line 186: the choice of times points is evidently different for mortality/infection measures and gene expression analyses...why?

lines 194-195: why is this relevant? the mapping rate as far as I understand shouldn't have anything to do with the biology of the host or parasite

line 204: "However" is the wrong word here, as this sentence is not in contrast with the previous statement

lines 210-212: this sentence seems like an explanation for the above statement...the two should be linked better

lines 224-226: have the authors checked also different time points? if so, these should be reported in the figure

line 230: "confirmed" should read "confirm"...however it's not a great start for a discussion section

lines 264-267: this statement doesn't seem too robust from a scientifically point of view

lines 272-274: this goes against what the authors stated just above, that after 7 days the parasite starts proliferating...

line 323: define "TO"

line 324: is this mRNA referred to TO?

lines 391-392: specify how many individuals, more than two is too vague

lines 392-394: why the approach to quantify parasites is this different here from what was done in the other assay?

lines 396-397: are these one sample for each type of culture that was inoculated? this concept is never explained but appears in figures 2A&B and 3A&B

line 401: there are only 18 RNAseq samples uploaded in the online repository...there should be 26. Why are some missing?

line 402: part of the description of the analysis for the honeybee RNAseq data are a bit confusing....the honey bee genome is available and it has been so since 2005. Why didn't the authors

used tools like TopHat or STAR or similar (as they did for *L. passim*) to simply align the reads to the genome?

lines 425-427: why did the authors use Cufflinks or Cuffmerge to analyse honeybee RNAseq data instead of using SAMtools as they did for *L. passim*? it's very confusing that they used drastically different approaches to perform the same steps in two different organisms...also considering that one of the aims of the study is to do comparative analyses in the two systems!

line 436: can the authors explain this model better? what comparisons were performed exactly? It is quite difficult to understand how differentially expressed genes were obtained - and also how up/down-regulation was calculated...what was the reference in each case?

We like to thank three reviewers for the time and effort to review our manuscript first. Here are our detailed responses to the points raised.

Reviewer #1

Line 44: The decline of number of bee hives: please use a references, for instance Pots et al., 2010 Journal of Apicultural Research.

Authors: We have added several references including above in the revised manuscript (Line 41-42).

Line 47 and 49: References Evans and Schwarz 2011 a and b are exactly the same (see reference list).

Authors: We have corrected this mistake in the revised manuscript (Line 552-553).

Line 115: do not mention the temperature here as it causes confusion (looks as if the bees were infected at 33°C; I think you mean that bees were kept at 33°C).

Authors: The sentence has been changed to “We also tested the survival rates of honey bees infected by *L. passim* and fed with 50 % (w/v) sucrose at 33 °C under laboratory condition.” in the revised manuscript (Line 112-113).

Lines 294-298: The paper of Blanchette et al., 1999 refers to *Leishmania* inside the macrophages, which is a situation completely different from the extracellular development of *Lotmaria* in bees. Only comparison with the promastigotes in the sandfly alimentary tract seems valid to me. So whole this paragraph is much too speculative.

Authors: Unfortunately, the functions of Amastin have been only characterized with *Leishmania* amastigotes but not promastigotes. We agree with the reviewer’s point and the part of paragraph has been changed to “*L. passim* appears to change its cell surface proteins during the infection cycle. Similar to *C. bombi* and *Crithidia expoeki* (Schmid-Hempel et al 2018), multiple Amastin genes appear to exist in *L. passim* genome and some are up-regulated at PI 12-27 (Supplementary Table 1); however, many are down-regulated and GP63 is instead up-regulated throughout the infection cycle. *Leishmania* GP63 was proposed to be essential for the promastigotes to attach to the insect gut wall (D’Avila - Levy et al. 2006, d’ Avila-Levy et al. 2014, de Assis et al. 2012). *Leptomonas* GP63 was suggested to have the same function (Pereira et al. 2009), which may also apply to *L. passim* GP63. Thus, switching Amastin to GP63 on the cell surface could be critical to establish and maintain the infection of *L. passim* in the honey bee hindgut.” in the revised manuscript (Line 274-284).

Lines 322: To my knowledge, gases like O₂ and CO₂ can easily cross the membranes of cells. This happens for instance in the capillaries of alveoli (lungs) and peripheral tissues of mammals, where oxygen easily crosses the endothelial cells. I would be very surprised if tight junctions between epithelial cells would have any influence on oxygen

uptake. Moreover, I do not fully understand your statement that *L. passim* would inhibit the oxygen supply from the basolateral side.

Authors: Due to the minor changes in the results obtained with new RNA-seq samples, this sentence has been deleted in the revised manuscript.

Line 339: Please explain better why the ‘*L. passim*-honey bee’ model is such an interesting model. In fact, I see many parallels with the ‘*Paenibacillus* larvae-honey bee larvae’ model: the pathogen can be cultured *in vitro*, can be easily administered, gene expression studies of the host (Cornman et al., 2013 PLOS ONE) and the pathogen (De Smet et al., 2014 PLOS ONE) have been studied. In fact, in the case of American foulbrood clinical signs are more obvious, effects on survival more dramatic and many virulence genes are known. Why would the *Lotmaria*-bee model be superior to this one?

Authors: We do not argue ‘honey bee-*L. passim*’ is the best model to study host-parasite/pathogen interaction. In fact, there are many examples including ‘honey bee-*P. larvae*’ model as the reviewer pointed out. We have therefore changed the sentence to “Honey bee-*L. passim* should represent a good model system to understand host-parasite interactions at three different layers, cell, organism, and population (colony).” in the revised manuscript (Line 327-329).

Line 386: universal bacteria: please explain what this is.

Authors: Universal bacteria represent all bacterial species. This has been mentioned in the revised manuscript (Line 372).

Line 393: I regret that only one reference gene was used for the qRT-PCR. In fact, you should have determined how many reference genes are required and which are the most stable ones, as done for the first time in honey bee research by Scharlaken et al., 2008 Journal of Insect Science.

Authors: To prepare the new RNA-seq samples, we quantified *L. passim* using primers to detect the 18S rRNA and the reference was honey bee 18S rRNA. We agree with the reviewer’s point and 18S rRNA should be one of the most stable reference genes. This has been mentioned in the revised manuscript (Line 379-381). The primer sequences are listed in Supplementary Table 9.

Reviewer #2

The Introduction and the Discussion sections are extraordinary well written. However, Material & Methods and the Result sections seem to have been written by inexperienced researcher and must be significantly corrected (adapted for publishing in scientific paper).

Authors: We have extensively modified the Materials & Methods as well as Result sections of the revised manuscript. The writing was also checked by a native English

colleague in our Department. We have done our best; however, understand there is also space to improve English writing.

I strongly suggest mentioning the overall number of analyzed genes in both the Abstract and the M&M section. In Supplementary Table 11 there are data for only 22 genes although the obtained results indicate that much more genes were included in the study. I also suggest authors to provide the complete list of all analyzed genes (together with primers used) in one Table. There are 31 specific comments/suggestions I inserted directly in MS in order to help authors to correct the MS.

Authors: All *L. passim* and honey bee genes described in the manuscript were identified as differentially expressed genes by bioinformatic analysis of 30 *L. passim* and 48 honey bee transcriptomes. Only honey bee Vitellogenin mRNA was directly analyzed by qRT-PCR. Due to the large number of genes identified, it will take very long time to verify all of the bioinformatic results by qRT-PCR. We have tested 15 *L. passim* mRNAs by qRT-PCR and the results are basically the same as those obtained by the bioinformatic analysis.

All of the reviewer's comments/suggestions have been reflected in the revised manuscript except the followings.

Line 115-117: This sentence has remained in the revised manuscript since it is interpretation of the results shown.

Line 119-122: We think these sentences help the reader to understand the aim and outline of experiments; however, we have made them quite brief in the revised manuscript.

Line 128-130: The level of parasite infection in each RNA-seq sample was tested by qRT-PCR as described in the Materials and Methods section (Line 379-381). Furthermore, the mapping rate of RNA-seq reads to *L. passim* genome also indicates the parasite load in each sample (Supplementary Table 1).

Line 205-208: The sentence has remained in the revised manuscript to briefly explain the aim of experiments. However, the order of two sentences has been changed.

Line 381-385: We designed the primers to detect *L. passim* 18S rRNA by qRT-PCR in this study (Supplementary Table 9).

In spite of high-quality data obtained, the presentation does not meet the criteria expected for scientific journal. In fact, Material & Methods and the Result sections seem to have been written by inexperienced researcher and must be significantly corrected (adapted for publishing in scientific paper). Reporting of data and methodology is sufficiently detailed, but the writing style is not adequate.

Authors: Please refer to our responses above.

Overall number of analyzed genes is not clear, so I strongly suggest mentioning that number in both the Abstract and the M&M section. In Supplementary Table 11 there are data for only 22 genes although the obtained results indicate that much more genes were included in the study (e.g. see the sentence: 'Pair-wise comparisons between cultured parasites and honey bee-infecting parasites indicated that 961, 415, 582 genes were up-regulated and 451, 1069, 658, 832 genes were down-regulated at PI 7, 12, 20, and 27, respectively' lines 142-144 in MS as well as Supplementary Tables 1-9). Thus, I suggest authors to provide the list of all analysed genes (together with primers used).

Authors: Please refer to our responses above.

The manuscript reference previous literature appropriately. Nevertheless, some additional references should be included as I indicated directly in MS.

Authors: The suggested references have been added in the revised manuscript.

Sections M&M and Results must be corrected (in accordance to my comments/suggestions inserted directly in MS).

Authors: Please refer to our responses above.

Reviewer #3 (Remarks to the Author):

In this study, Liu et al. approach an interesting host-parasite system (the honeybee *Apis mellifera* and a trypanosome gut parasite) trying to understand the intricate array of interactions between the two organisms from multiple points of view. I much appreciate the comprehensive approach, aimed at considering the effect of the parasite on host survival, on the microbial flora associated with the host gut and host gene expression over time during the infection. Furthermore, the authors planned to look at gene expression in the parasite too, to see whether it follows the fluctuations observed in the host. This is quite a novel and challenging approach that must be commended. However, probably for the high challenges associated with such a complex and multi-faceted study, the authors fail in my opinion to provide any strong evidence for any of these aspects.

My major concern is about the design of the RNAseq studies. Sample size is too low there – only two samples for experimental group – and this cannot be accepted as it doesn't really provide any robust scientific significance. Three would be the minimum to calculate any sort of average/median of gene expression data, and even this would be too low when considering individual biological replicates that might vary for so many factors. Nowadays for individual replicates, normally scientist aim to have at least 6-8 samples per group and this is so far from what the authors have here. In FIG. 3A for example, when the two samples in the infPI12 group are so much separated in the PCA analysis, what can be inferred? Furthermore, also the coverage for the RNAseq approach is so low (4GB) compared to what is normally the case these days (20GB or more) that it makes it difficult to consider these results as robust and significant.

Even though sample size is not a major issue for the other analyses (as it is much more reasonable than in the RNAseq study) the authors are not always clear about it through

the manuscript. The reader can infer it from the figures of Vg expression, for example, or the quantification of *L. passim* and bacteria...but it would help if authors could clearly state it in the format $N = XX$ in the text (results for examples) and in the figure or figure legend. For mortality, sample size is huge but this doesn't appear from the main text – I realised it only after reading the reporting summary.

There are also issues associated with the presentation of the gene expression data and the description of the approaches of analysis that makes it a bit confusing to understand what the authors have done – see points below.

On a positive note, the paper is well written overall, indicating a deep knowledge of the field and thorough research of the literature. The discussion is very comprehensive and the authors have made a good effort to analyse the gene expression results in the context of the biology of the host-parasite system and the previous literature on the same system or similar host-parasite associations. It is unfortunate that the RNAseq study is so weak, and the rest of the results, though possibly interesting, disappear through the manuscript as so much emphasis is given to the gene expression analyses.

Authors: We increased the number of replicates from two to six for each RNA-seq sample in the revised manuscript. To collect these samples, we had to use the honey bee colony different from the one previously used. Thus, *L. passim* was infected to honey bees with the different genetic background and perhaps gut microbiota as well. This likely resulted in the lower level of infection with the new samples compared to the previous ones. For example, the average mapping rate of RNA-seq reads to *L. passim* genome at PI 27 was 74 % for the previous samples; however, it was reduced to 46.7 % for the new samples (Supplementary Table 1). Clustering analysis of 30 *L. passim* transcriptomes based on the similarity showed some of them at the same time point do not cluster together (Supplementary Figure 1A). This was more pronounced with 48 honey bee transcriptomes as expected (Supplementary Figure 1B). This is due to the large heterogeneity in individual honey bee transcriptome and physiological state as previously reported. Nevertheless, we used all samples for the bioinformatic analysis without eliminating these outliers. Despite above differences between the new and previous RNA-seq samples, we basically obtained the similar results and the major conclusions remain the same in the revised manuscript.

We agree that the deep sequencing (20 GB) is better particularly for the statistical analysis of very rare transcripts. However, this was not possible for us and the problem associated with 6 GB sequencing has been mentioned in the revised manuscript (Line 384-386).

The sample sizes have been described in the figure legends of the revised manuscript as suggested (Line 778, 784, 789, 805-806).

line 76: rephrase

Authors: The sentence has been changed to “Several genes in immune signaling pathways, such as Spatzle, MyD88, Dorsal, Defensin, Hymenoptaecin and Apidaecin are also up-regulated during the early stage of *C. bombi* infection in bumble bee

(Riddell et al. 2011, Riddell et al. 2014, Schlüns et al. 2010).” in the revised manuscript (Line 72-75).

line 77: Drosal = Dorsal

Authors: This has been corrected as above.

lines 137-138: this is a confusing sentence. Who are "these" and what are the "above results"? please clarify

Authors: The sentence has been changed to “These results were consistent with Fig. 1A showing that parasite population increases 8 days after infection under laboratory condition.” in the revised manuscript (Line 135-137).

lines 141-142: what does “and it was an outgroup to the other time points” mean?

Authors: This has been deleted in the revised manuscript.

FIG. 2&3 C and D: the way how gene expression results are presented in these sections of the two figures and also in the associated main text is a very confusing. Why did the authors decided to present gene expression data with convoluted overlapping analyses represented by confusing Venn Diagrams? I understand the overlaps to identify common genes across time points – but this could have been simply added to the supp mat and discussed in the result section. As it is a time-course study, it would be more indicative to show gene expression data with line graphs indicating numbers of genes up or down-regulated that go increase or decrease over time in the host and in the parasite. also, what is the up or down-regulation referred to? in the host, my assumption would be that the comparisons are done between infected and control individuals at each time point - so up/down refer to patterns in infected bees with respect to controls...is this true? It’s impossible to understand this from the results or the methods. And what about the parasite? are they all compared to parasites in culture? A or B or together? or did the authors followed a completely different approach? this must be explained.

Authors: We have kept the Venn diagrams in Figures 2 and 3 of the revised manuscript. The numbers of up- and down-regulated genes at the different stages of infection have been described in the text (Line 137-140 and 192-195) and these should be enough to understand the general trend of changes in the gene expression. We believe it is more informative to show the numbers of genes up- and down-regulated at the specific stages of infection or throughout the infection period.

To identify *L. passim* DEGs, comparison was made between the cultured parasites and the parasites at the different stage of infection in honey bees. This has been described in the text (Line 137-140 and 421-423) and the lists of DEGs with the results of GO term enrichment analysis are shown in Supplementary Table 2. Furthermore, the DEGs between the different stages of infection (PI 7 vs PI 12, PI 12 vs PI 20, and PI 20 vs PI 27) were also identified and the lists of DEGs with the results of GO term enrichment analysis are shown in Supplementary Table 7. This has been mentioned in the text (Line

167-170). The honey bee DEGs were identified between the infected and uninfected control honey bees at the same age/(stage of infection). This has been described in the revised manuscript (Line 182-184, 192-195, and 423-425).

lines 169-173: this should be stated much earlier in the manuscript to provide a rationale for these comparisons...which otherwise are difficult to understand at first for people who are not expert in the field

Authors: This has been moved to the Discussion section of the revised manuscript as suggested by the reviewer #2 (Line 237-284).

line 186: the choice of time points is evidently different for mortality/infection measures and gene expression analyses...why?

Authors: The survival of honey bees were recorded every day. The samples to measure the levels of parasite infection under laboratory condition were collected at 1, 3, 8, 15, and 22 days after the infection. The RNA-seq samples were collected at 7, 12, 20, and 27 days after the infection. The schedule of sample collection was slightly changed depending on the start date of parasite infection.

lines 194-195: why is this relevant? the mapping rate as far as I understand shouldn't have anything to do with the biology of the host or parasite

Authors: Each RNA-seq sample of the parasite-infected honey bee contains the transcript reads derived from both parasite and honey bee. Thus, the mapping rate of RNA-seq reads to either *L. passim* or honey bee genome indicates the level of infection in each parasite-infected honey bee.

line 204: "However" is the wrong word here, as this sentence is not in contrast with the previous statement

Authors: This sentence has been deleted in the revised manuscript.

lines 210-212: this sentence seems like an explanation for the above statement...the two should be linked better

Authors: This sentence has been deleted in the revised manuscript.

lines 224-226: have the authors checked also different time points? if so, these should be reported in the figure

Authors: We analyzed the samples only at 20 days after the parasite infection and this has been mentioned in the figure legend (Line 806).

line 230: "confirmed" should read "confirm"...however it's not a great start for a discussion section

Authors: The sentence has been changed to “We found that ingested *L. passim* reaches the honey bee hindgut and colonizes it as previously reported (Schwarz et al. 2015).” in the revised manuscript (Line 214-215).

lines 264-267: this statement doesn't seem too robust from a scientifically point of view

Authors: This sentence has been deleted in the revised manuscript.

lines 272-274: this goes against what the authors stated just above, that after 7 days the parasite starts proliferating...

Authors: This sentence has been deleted in the revised manuscript.

line 323: define “TO”

Authors: “TO” is defined earlier at Line 199 of the revised manuscript.

line 324: is this mRNA referred to TO?

Authors: Yes and the sentence has been changed to “TO was shown to be involved in controlling the feeding behavior and nutritional status of fruit fly by binding juvenile hormone and the mRNA expression is highly stimulated in the head and gut tissues by starvation (Sarov-Blat et al. 2000, So et al. 2000).” in the revised manuscript (Line 304-307).

lines 391-392: specify how many individuals, more than two is too vague

Authors: Total RNA was extracted from 12 individual honey bees at each time point. This has been described in the revised manuscript (Line 378-379).

lines 392-394: why the approach to quantify parasites is this different here from what was done in the other assay?

Authors: In Fig. 1A, *L. passim* was quantified by qPCR targeted to the genomic region for ITS of rRNA using total DNA isolated from the parasite-infected honey bee as the template. However, total RNA was extracted for the RNA-seq sample and thus qRT-PCR targeted to *L. passim* 18S rRNA was used to roughly estimate the level of parasite infection in each RNA-seq sample. This has been described in the revised manuscript (Line 379-381).

lines 396-397: are these one sample for each type of culture that was inoculated? this concept is never explained but appears in figures 2A&B and 3A&B

Authors: Each RNA-seq sample was prepared with single *L. passim*-infected or uninfected control honey bee. We also prepared six RNA-seq samples with six independently cultured *L. passim*. These have been described in the revised manuscript (Line 376-388).

line 401: there are only 18 RNAseq samples uploaded in the online repository...there should be 26. Why are some missing?

Authors: The new RNA-seq data (PRJNA587465) were deposited in NCBI and there are 54 samples in total. There are 6 samples of cultured *L. passim*, 24 samples of the parasite-infected honey bees, and 24 samples of the control uninfected honey bees (Line 386-388).

line 402: part of the description of the analysis for the honeybee RNAseq data are a bit confusing....the honey bee genome is available and it has been so since 2005. Why didn't the authors use tools like TopHat or STAR or similar (as they did for *L. passim*) to simply align the reads to the genome?

lines 425-427: why did the authors use Cufflinks or Cuffmerge to analyse honeybee RNAseq data instead of using SAMtools as they did for *L. passim*? it's very confusing that they used drastically different approaches to perform the same steps in two different organisms...also considering that one of the aims of the study is to do comparative analyses in the two systems!

Authors: We agree two previous paragraphs explaining the analysis of RNA-seq data were indeed quite confusing. In the new subsection "Gene annotation for *L. passim* genome and transcriptome sequences" of the revised manuscript, we described how genes were annotated in both genomic and transcriptome sequences of *L. passim*. This was necessary since the genome sequence of *L. passim* available in NCBI does not contain the information for gene annotation. The gene-annotated GFF3 file and the protein coding sequences were used for the bioinformatic analyses. In the new subsection "Mapping of RNA-seq reads to honey bee and *L. passim* genomes as well as the hierarchical analysis of RNA-seq samples" of the revised manuscript, we described how we determined the mapping rates of clean RNA-seq reads to honey bee and *L. passim* genomes using HISAT2 for both cases. We also explained the methods to generate heat maps to show the results of hierarchical analysis of 30 *L. passim* and 48 honey bee transcriptomes using Pearson correlation.

line 436: can the authors explain this model better? what comparisons were performed exactly? It is quite difficult to understand how differentially expressed genes were obtained - and also how up/down-regulation was calculated...what was the reference in each case?

Authors: Please refer to our previous responses regarding above points.

Reviewers' comments:

Reviewer #1 (Remarks to the Author):

The authors have addressed most of the comments raised. Only the issue of the reference genes remained unanswered.

Reviewer #2 (Remarks to the Author):

Since I have already evaluated and commented this paper, this time I checked how the authors corrected it and I concluded that they had substantially improved it according to given recommendations. In some cases they decided not to make suggested changes, but they provided a satisfactory explanation. Thus, I think the work can be accepted in its current form.

Reviewer #3 (Remarks to the Author):

As this is the second time I review the study by Liu et al. I mainly focused on the aspect that was the major source of concern in my first evaluation: that is, the design and execution of the RNAseq experiment. I must say that it is commendable that the authors decided and managed to increase the sample size for the experiment, from $N = 2$ (totally insufficient to gather any statistical significance) to $N = 6$, which is an acceptable sample size for RNAseq experiments. Considering the time required to collect the samples and process them for molecular work, and also the costs associated with RNAseq this has to be praised.

However, I find it worrisome that in the analyses the authors combined the first set of samples with the second one, thus introducing a long list of external factors that could introduced biases in the data and therefore in the analysis and interpretation of the results. Some of these factors are promptly commented upon by the authors themselves in their rebuttal (without appearing though in the manuscript itself): for example, different strains of the parasites from year 1 to year 2 that resulted in different infectivity levels, and also the use of different colonies that possibly introduced significant differences in the susceptibility levels of the insect host. Gene expression is very sensitive to any possible factor that affect the organism internally or externally, and therefore I wouldn't be surprised that genes of both the host and the parasite respond more to the batch effect (i.e. year 1 vs. year 2) than to the infection itself. On top of this, it is generally not recommended to clump together RNAseq dataset from different experiments (different years) and analyse them together – unless it is possible to correct for this batch effect in the analysis. RNAseq is an extremely sensitive approach that responds to even subtle variations occurring during preparation of the samples and sequencing. It seems that by using samples from two different experiments basically the authors might have introduced some serious technical biases. Have the authors performed any statistical analysis to check whether there is any batch effect of this sort? I haven't found any reference to this in the manuscript. For example, I'm wondering whether the samples would cluster more based on the year of processing instead of infection status? Also, a possible batch effect could explain why there is such huge difference between the numbers of genes differentially expressed in the parasite in the first version of the manuscript vs. the second. These numbers often vary 2-3 folds and in one case even 5 folds, from 415 genes to 2531 genes upregulated in the parasite at PI 20...I'm not even sure how many genes are in the parasite genomes but 2531 seems a lot and above all it is a very different number from 415! This difference is not particularly reassuring and I find it hard to believe that the results of this second version of the manuscript are consistent with the previous one, as the authors claim.

Also, I have noticed that some of the p-values – for example those in supplementary table 77119 – are incredibly small (for example $5.04E-247$) even after correction for multiple comparisons. This

seems to be a bit extreme, normally such kind of p-values in my experience denote that the authors did not select the right statistical test or did not control for dispersion of data points properly. I am not sure of what should be recommended in this case. Have the authors tried to analyse the new dataset alone and see whether there is any consistence with the previous analysis? 4 replicates per group is not as good as 6 but if there is more consistence among the replicates the outcome might be more meaningful.

Reviewers' comments:

Reviewer #1 (Remarks to the Author):

The authors have addressed most of the comments raised. Only the issue of the reference genes remained unanswered.

Authors: As described in the revised manuscript (Line 379-381), we used honey bee 18S rRNA as the reference gene since it is one of the most abundant and thus stable RNA in the cells. The primer sequences are listed in Supplementary Table 9.

Reviewer #2 (Remarks to the Author):

Since I have already evaluated and commented this paper, this time I checked how the authors corrected it and I concluded that they had substantially improved it according to given recommendations. In some cases they decided not to make suggested changes, but they provided a satisfactory explanation. Thus, I think the work can be accepted in its current form.

Authors: Many thanks for the comment.

Reviewer #3 (Remarks to the Author):

As this is the second time I review the study by Liu et al. I mainly focused on the aspect that was the major source of concern in my first evaluation: that is, the design and execution of the RNAseq experiment. I must say that it is commendable that the authors decided and managed to increase the sample size for the experiment, from $N = 2$ (totally insufficient to gather any statistical significance) to $N = 6$, which is an acceptable sample size for RNAseq experiments. Considering the time required to collect the samples and process them for molecular work, and also the costs associated with RNAseq this has to be praised.

However, I find it worrisome that in the analyses the authors combined the first set of samples with the second one, thus introducing a long list of external factors that could introduced biases in the data and therefore in the analysis and interpretation of the results. Some of these factors are promptly commented upon by the authors themselves in their rebuttal (without appearing though in the manuscript itself): for example, different strains of the parasites from year 1 to year 2 that resulted in different infectivity levels, and also the use of different colonies that possibly introduced significant differences in the susceptibility levels of the insect host. Gene expression is very sensitive to any possible factor that affect the organism internally or externally, and therefore I wouldn't be surprised that genes of both the host and the parasite respond more to the batch effect (i.e. year 1 vs. year 2) than to the infection itself. On top of this, it is generally not

recommended to clump together RNAseq dataset from different experiments (different years) and analyse them together – unless it is possible to correct for this batch effect in the analysis. RNAseq is an extremely sensitive approach that responds to even subtle variations occurring during preparation of the samples and sequencing. It seems that by

using samples from two different experiments basically the authors might have introduced some serious technical biases. Have the authors performed any statistical analysis to check whether there is any batch effect of this sort? I haven't found any reference to this in the manuscript.

For example, I'm wondering whether the samples would cluster more based on the year of processing instead of infection status? Also, a possible batch effect could explain why there is such huge difference between the numbers of genes differentially expressed in the parasite in the first version of the manuscript vs. the second. These numbers often vary 2-3 folds and in one case even 5 folds, from 415 genes to 2531 genes upregulated in the parasite at PI 20...I'm not even sure how many genes are in the parasite genomes but 2531 seems a lot and above all it is a very different number from 415! This difference is not particularly reassuring and I find it hard to believe that the results of this second version of the manuscript are consistent with the previous one, as the authors claim.

Also, I have noticed that some of the p-values – for example those in supplementary table 77119 – are incredibly small (for example 5.04E-247) even after correction for multiple comparisons. This seems to be a bit extreme, normally such kind of p-values in my experience denote that the authors did not select the right statistical test or did not control for dispersion of data points properly.

I am not sure of what should be recommended in this case. Have the authors tried to analyse the new dataset alone and see whether there is any consistence with the previous analysis? 4 replicates per group is not as good as 6 but if there is more consistence among the replicates the outcome might be more meaningful.

Authors: For the revision of manuscript, we used **six new** RNA samples of cultured *L. passim*, the parasite-infected honey bee hindguts, and the control (uninfected) honey bee hindguts at each of 7, 12, 20, and 27 days after the infection for RNA-seq and the following bioinformatic analysis. These were described in the revised manuscript (Line 376-388). As we explained in the previous rebuttal letter, *L. passim* was infected to honey bees of the colony different from the one used in the original manuscript. Furthermore, we obtained 4Gb of clean reads for each RNA-seq sample using Illumina HiSeq-4000 platform in the original manuscript (Accession number: PRJNA510495); however, we used BGISEQ-500 platform to increase the sequencing depth to 6Gb in the revised manuscript (Accession number: PRJNA587465). Therefore, we, of course, **did not** combine above two different sets of RNA-seq data for the bioinformatic analysis. Mapping rates of **new** RNA-seq samples (54 in total) to either honey bee or *L. passim* genome are shown in Supplementary Table 1 and the results of hierarchical clustering of the **new** honey bee and parasite transcriptomes are also shown in Supplementary Figure 1 of the revised manuscript. The results obtained in the original and revised manuscripts were based on the bioinformatic analysis of the independent sets of RNA-seq samples (2 vs 6 samples). Nevertheless, the major conclusions were basically similar.

As shown in Supplementary Table 2 of the revised manuscript, it is true that the numbers of DEGs significantly increase with lower P-values compared to those in the original manuscript. We have repeated the bioinformatic analysis to identify the DEGs and obtained the same results. The results of GO-term enrichment analysis of the DEGs

were very similar between the original and revised manuscripts, suggesting that genes with the same functions were selected out in both cases. Given the difference of environment between the culture medium under normoxia and anaerobic honey bee hindgut, it is not surprising to find many genes change the expression. We believe these results were obtained by increasing the number of RNA-seq samples with more statistical power and thank the reviewer for the important suggestion.